# Bovine mastitis epidemiology: Prevalence, risk factors, control program gaps and biosecurity recommendations to improve animal health in the Rwandan smallholder dairy farms

Blaise Iraguha[1,2]*, Methode Ngabo Gasana[3], Jean Pierre M. Mpatswenumugabo[4]

1 Rwanda Dairy Development Project (RDDP), Kigali, Rwanda, 2 College of Veterinary Medicine, University of Illinois at Urbana-Champaign, Urbana, Illinois, United States of America, 3 Rwanda Agriculture and Animal Resources Development Board, Kigali, Rwanda, 4 Department of Veterinary Medicine, University of Rwanda, Nyagatare, Rwanda

* irablais2000@gmail.com

## Abstract

Bovine mastitis remains a significant challenge to dairy health management worldwide, with substantial economic and public health implications. In Rwanda, where traditional dairy farming is crucial for household livelihoods and the national economy, mastitis reduces milk yield and increases the risk for bacterial contamination, posing serious food safety concerns. This study, conducted in Rwanda's North-Western region from January 2024 to April 2024, aimed to identify key mastitis risk factors, evaluate existing control gaps, and propose evidence-based interventions. This cross-sectional study collected data from 411 smallholder dairy farms in Rwanda, assessing one lactating cow per farm through clinical examination, California Mastitis Test, and structured farmer questionnaires on management and hygiene practices. Logistic regression analysis in R identified significant cow-related and farm-level risk factors, providing a basis for targeted mastitis control and biosecurity recommendations. The overall mastitis prevalence was 60.06% (95% CI: 53.57–66.55), with subclinical cases alone accounting for 56.9%. Poor cow hygiene emerged as the strongest predictor (OR = 85.47, 95% CI: 27.18–268.74; p<0.001). Other associated factors included exotic pure breeds, late lactation stages, and multiparity. External contributors included inadequate milking practices and limited veterinary access. In zero-grazing systems, poor housing drainage (OR = 109, 95% CI: 26.46–507.18; p<0.001) and infrequent bedding changes (OR = 6.36, 95% CI: 3.38–12.78; p=0.014) significantly increased mastitis risk. Identified gaps included lack of farmer knowledge, insufficient access to affordable mastitis control supplies (e.g., disinfectants such as iodine or chlorine dioxide), inappropriate mastitis treatments, poor farm biosecurity, and inefficient quality control in the milk marketing chain. Strengthening farm biosecurity, implementing a national mastitis control program, and enhancing veterinary extension are essential to reduce mastitis and improve milk safety.

**Data availability statement:** All relevant data are available in the Zenodo repository at: https://doi.org/10.5281/zenodo.17873853.

**Funding:** The author(s) received no specific funding for this work.

**Competing interests:** The authors declare no conflicts of interest.

Coordinated stakeholder action is vital for sustainable dairy development and public health.

## Introduction

In recent decades, the health status of dairy cows has significantly improved as a result of global enhanced management practices and disease-preventive measures implementation [1]. However, despite these advancements, bovine mastitis remains a major health concern in dairy cows. It is a multifactorial and complex disease characterized by inflammation of the milk-producing parenchyma of the udder and elevated somatic cell counts. [2]. This results in considerable economic losses mainly associated with reduced milk production, altered milk quality, significant treatment costs, milk discards during treatment, culling and occasional deaths among dairy producers [3,4]. Thus, early diagnosis of mastitis constitutes crucial information as prevention and control measures are tailored based on where the pathogens originate from [5].

Generally, the incidence of mastitis varies by farms, regions, or countries. Mastitis factors involve mainly the characteristics of the cows, farming systems, milking practices, environmental conditions and control or management practices [6]. Factors such as age parity, breed, and stage of lactation are significant cow's characteristics closely related to mastitis outbreaks [7,8]. Teat end conditions, udder conformation, nutritional status, and body condition score were also observed as mastitis predisposing factors [7,9]. The significant association between mastitis occurrence and different production systems has also been recognized with the variability in isolated mastitis pathogens [10,11].

Environmental factors such as climatic conditions including humidity, rain and temperature affect cow's cleanliness and/or housing areas resulting in increased mastitis outbreaks in dairy cattle [10]. Environmental pathogens usually enter the udder through teat canals mainly after milking or in poor udder hygienic conditions. Hence, housing conditions, milking areas, and milking practices are crucial factors associated with mastitis occurrence [1]. Moreover, access to fresh and clean water is very important feature for animal health improvement not only for supporting the immune system, milk production and maintaining proper udder function but also for maintaining good hygienic practices, reducing the risk of udder infection and mastitis transmission [6]. The influence of grazing areas including pastures, cowsheds, and bedding management in confinement constitute important mastitis factors. This is attributed to the likelihood of the dairy cow's tendency to lay down after milking, bringing their udder and teats in contact with bedding or dusty environment [6]. Thus, the differences in bedding size, shape, and replacements further influence the occurrence of environmental mastitis mainly in small-scale dairy farming [12]. Most importantly, the selection of bedding materials depends primarily on their availability and costs, with dairy producers choosing the cost-effective and accessible options [6]. Additionally, bacterial control by effective use of teat dips plays a crucial role in reducing clinical mastitis incidence when applied appropriately and regularly. This was demonstrated by a reduced 54% major mastitis intramammary infection following the

application of teats pre-dipping [5]. Proper application of suitable post-milking teat dips is highly recommended to prevent the transmission of contagious mastitis such as *Mycoplasma bovis, Streptococcus agalactiae* and *Staphylococcus aureus* [5]. Additionally, appropriate milking procedures involving wearing gloves, fast milking, stripping and thoroughly drying of the teats before milking using clean absorbents or paper towels is crucial [9].

While mastitis has been widely studied in high-income countries, there is limited evidence from smallholder systems such as in sub-Saharan Africa particularly on how different management practices influence disease dynamics. Given mastitis is an endemic disease, farmers in most developed countries apply control programs to keep its prevalence below 5% [9] and the success of these programs are based on a comprehensive understanding of mastitis dynamics and system-specific challenges. Such knowledge is essential in small-scale farming to minimize the detrimental effects of mastitis on the dairy industry and is currently missing in the Rwandan dairy chain. Therefore, this cross-sectional study aimed primarily (1) to investigate mastitis prevalence and associated risk factors among lactating cows, (2) identify mastitis control gaps in the current grazing systems, and (3) propose actionable solutions for improving mastitis control and prevention strategies.

## Materials and methods

### Ethics declaration

This study did not involve the collection of human or animal biological samples. Formal IRB review was not required because the study focused on farmer interviews and animal observations without collecting biological or sensitive personal data. Ethical approval was obtained from the relevant administrative authorities at the district level in Nyabihu and Musanze, Rwanda.

As many participants were unable to read or write, verbal informed consent was obtained prior to each animal clinical assessment and interview, and the process was witnessed and documented by a local veterinary or extension officer. The study complied with Rwanda's national research ethics guidelines and the principles of the Declaration of Helsinki.

### Study area

This study was conducted in Nyabihu (1°39′9.90″S; 29°30′24.62″E) and Musanze (1°30′6.94″S; 29°37′59.75″E) districts with annual mean temperature and rainfall ranging around 18.1°C (1400 mm) and 15°C (1100 mm); respectively [13]. Both districts undergo two dry seasons and two rainy seasons annually. The moderate rainfall occurs in October and November; the heaviest rainfall falls in April and May while the short dry season stretches from January to mid-March. This area was chosen as a prominent hub for raw milk production and processing accommodating 8 active milk collection centers, over 60% of the Rwandan cheese processors, and the major milk processing plant, Mukamira Dairy [13]. These districts also represent both grazing systems practiced in Rwanda: (1) open grazing, where cows wander at the pastures freely during the day and night, and (2) zero-grazing, where cattle are entirely enclosed and fed within a kraal [14]. Additionally, 51.6% of the total population in this area own cows [13].

### Study design and sample size determination

A cross-sectional study was conducted using stratified random sampling to target the two main dairy grazing systems in Rwanda: open grazing and zero-grazing. Using Yamane's formula [15] at 95% confidence interval, a total of 411 lactating cows were randomly selected: 281 from open grazing systems and 130 from zero-grazing systems. Field data were collected from 5 January 2024–30 April 2024. One lactating cow per farm was selected, chosen at random if more than one was present. The sampled cows included approximately 8% pure Friesians and 92% undefined crossbreeds, primarily involving dairy breeds mixed with indigenous Ankole cattle. Each farm owner or manager was interviewed using a structured questionnaire to obtain data on management practices. In total, 411 farms were included, with 206 from Nyabihu District and 205 from Musanze District.

The one-to-one pairing of each cow with its respective farm enabled a direct linkage between mastitis outcomes and both cow and farm-level risk factors. This design enabled a robust evaluation of how management practices, housing conditions, and breed types influence mastitis occurrence.

## Mastitis screening procedures and cow-related factors assessment

Data from each individual lactating cow was gathered during on-site visits. Each enrolled cow underwent a series of assessments, including physical examinations, subclinical mastitis testing and intrinsic factors recording. Clinical examination included a thorough assessment of the udder and teats including palpation to identify signs of inflammation such as swelling, pain, fibrosis, visible injuries, tissue atrophy, infestation by ticks, and swelling of the supramammary lymph nodes. Mammary quarters and teats were inspected for consistency, size, and milk secretion. Then, California Mastitis Test (CMT) was used to screen subclinical mastitis based on the degree of coagulation [16]. Prior to milk sampling, the udder and teats were cleaned by hand washing using clean water. Following washing, fore-stripping of four to five streams of milk was applied and then dipped the teats in 0.5% iodine solution for approximately 30 seconds and subsequently dried using a clean towel.

A direct squirt of milk from each individual quarter was placed into the four shallows of the CMT paddle. An equal quantity of the CMT reagent was then added and a gentle circular motion in horizontal plane applied to facilitate the testing process and interpreted the results according to the degree of coagulation graded as "–" (negative), "+" (Slightly positive), "++" (Moderately positive), and "+++" (Highly positive) [16]. Information including grazing system (open or zero-grazing), breed (pure exotic or crossbreed), lactation stages (early, middle, or late), parity (primiparous, multiparous, or grand multiparous), age (young or old), presence of teat ulcerations and cracks, overall cow cleanliness as well as mastitis results were recorded using cow-specific questionnaires. Cow cleanliness was assessed using a scoring chart rating as (1) not dirty (any trace of mud exists on the udder and the rest parts), (2) slightly dirty (a very small traces of mud fill the udder and other part remaining clean), (3) fairly or moderately dirty (a part of udder and small scale of back, tail and belly muddy) and (4) very dirty (the entire udder, back legs, tail, and belly are muddy filled) [17].

## Extrinsic mastitis risk factors assessment

Through field visits and using structured questionnaires, 411 smallholder dairy farmers were interviewed to assess mastitis related extrinsic factors. The questionnaire comprised dichotomous, multiple-choice, and open-ended questions and covered 8 subparts including social demographic information, milking practices, hygienic milk handling, mastitis control practices, farming and waste management practices, environmental factors, access to services and other farm biosecurity questions (Table 1).

## Data analysis

Collected data were recorded in Excel spreadsheets, encompassing a comprehensive range of variables such as mastitis cases, intrinsic factors, and extrinsic factors. Data analysis was performed using R software version 4.4. Descriptive statistical analysis assessed the sociodemographic characteristics and mastitis prevalence, and results were summarized as counts and percentages with corresponding 95% confidence intervals (CIs).

Fisher's exact and chi-square tests were used to compare on-farm risk factors associated with mastitis (SCM and CM positivity). A multinomial regression model was developed for each independent variable: intrinsic and cow related factors, on-farm milking practices, on-farm mastitis control and farming practices and lastly environmental and other external influencing factors. Model fitness was checked using the Akaike Information Criterion (AIC) and Bayesian Information Criterion (BIC) to compare the models.

Logistic regression model provided critical insights into the key mastitis risk factors and current control gaps, offering a foundation for targeted preventive interventions.

**Table 1. Summary questionnaire administered to individual dairy farmers to assess mastitis risk factors.**

| Variables | Categories |
|---|---|
| Milking techniques | Hand milking/ milking machine |
| Milking length | Less 7 mins, between 7–15 mins, over 15 mins |
| Do you let calf suckle before, between and after milking? | Yes/ No |
| Grazing type | Open grazing/ zero grazing |
| Presence of cowshed | Yes/ No |
| Presence of manure pit, compost, water harvesting system and presence of farm record keeping | Yes/ No |
| Is there adequate lighting in the milking shed? | Yes/ No |
| Presence of slope and well-drained cowshed | Yes/No |
| How frequent do you change bedding? | Everyday/ Less one week/ Between 1 week- two weeks/ Once a month or over |
| How long does it take a cow to lay down after milking? | More than an hour/ Between 30 mins – 1 hour/ Less than 30 mins |
| Do you check for clinical mastitis before milking (teat striping)? | Yes/ No |
| Frequency of subclinical mastitis screening. | Everyday / Once a week/ Once a month/ Never done |
| Who usually treats the animal (administers the medicine) during mastitis treatment? | Veterinarians / Myself/ Neighbor farmer |
| Teat dipping (pre and post dipping), dry cow therapy | Yes/ No |
| What do you do with milk from animals undergoing mastitis treatment using antibiotics? | Sold/ Drink at home / Withheld/Discarded/ Fed to calves/ Others (e.g., feed pigs) |
| Do you have clean water on the farm? | Yes/ No |
| How do you prepare the udder preparation before milking? | Clean with cold water only/ Clean the udder with warm water only/ Apply milking jelly/salve |
| How do milkers prepare before milking? | Warm water only/ Warm water and detergent/ Cold water only/ Cold water and detergent |
| Availability of private/public veterinarians | Yes/No |
| Access to training | Yes/ No |
| Use of molasse | Yes/ No |
| Frequency of spraying against vectors (ticks and mosquitoes) | Twice a week/ Once a week/ Over a week |

## Results

### Social demographic characteristics of respondents

Musanze district accounted for a higher proportion of zero-grazing farms. On average, a farmer owned two cows with differences between open and zero grazing. 55.2% of respondents were male. While most respondents had basic literacy skills and had completed primary school, their educational attainment was limited. Moreover, almost 86% did not receive any support from dairy development projects and only 20.9% were members of any dairy cooperatives (Table 2).

### Mastitis prevalence

Out of 411 milking cows screened for mastitis using California Mastitis Test (CMT), the overall cow-level mastitis prevalence was 60.06% (95% CI: 53.57, 66.55) with 56.9% (95% CI: 52.1, 61.7) subclinical mastitis and 3.16% (95%

**Table 2. Characterization respondent and herd information per grazing system (n = 411).**

| Variables | Categories | Total | Grazing systems | |
|---|---|---|---|---|
| | | | Open grazing | Zero-grazing |
| **Age** | Mean Age [IQR] years | 47.2 [23-80] | 47.2 [23-80] | 47.8 [23-72] |
| **Cow ownership** | Mean Cows [IQR]/ Farmer | 2 [1-20] | 4 [3-20] | 1 [1-4] |
| | | N (%) | N (%) | N (%) |
| **Location (district)** | Nyabihu | 206 (50.0) | 206 (73.3) | – |
| | Musanze | 205 (50.0) | 75 (26.7) | 130 (100) |
| **Sex** | Male | 227 (55.2) | 151 (53.7) | 76 (58.5) |
| | Female | 184 (44.8) | 130 (46.3) | 54 (41.5) |
| **Read and write** | Yes | 351 (85.5) | 235 (83.6) | 116 (89.2) |
| | No | 60 (14.5) | 46 (16.4) | 14 (10.8) |
| **Highest education level** | Primary | 271 (65.9) | 177 (63) | 94 (72.3) |
| | Secondary | 84 (20.4) | 61 (21.7) | 23 (17.7) |
| | College/Higher Education | 2 (0.1) | 2 (0.7) | – |
| | No formal education | 54 (13.2) | 41 (14.6) | 13 (10.0) |
| **Type of household** | Male-headed (husband & wife) | 337 (82.0) | 236 (84) | 101 (77.7) |
| | Female- headed (husband & wife) | 40 (10.0) | 20 (7.1) | 20 (15.4) |
| | Female-headed (Wife no husband) | 30 (7.2) | 23 (8.2) | 7 (5.4) |
| | Child-headed (No father and mother) | 4 (1) | 2 (0.7) | 2 (1.5) |
| **Previously assisted by development projects** | Yes | 59 (14) | 54 (19.2) | 5 (4.0) |
| | No | 352 (86) | 227 (80.8) | 125 (96.1) |
| **Member of a dairy cooperative** | Yes | 86 (20.9) | 80 (28.5) | 6 (4.6) |
| | No | 325 (79.1) | 201(71.5) | 124 (95.4) |

CI: 1.47–4.85) clinical mastitis. 565 out of 1639 [34.4% (95% CI: 32.1, 36.7)] functional examined quarters were positive for subclinical mastitis. 5/1644 quarters were blind while 6/411 cows exhibited changes in udder symmetry (Table 3).

## Mastitis risk factors

**Intrinsic and cow related predisposing factors.** The results for intrinsic and cow related mastitis predisposing factors are presented in Table 4. Mastitis prevalence was highly recorded in zero grazing than open grazing (OR = 2.93, 95% CI [1.81, 4.76], p < 0.001). Exotic pure breed cows exhibited higher odds of mastitis than crossbreed cows (OR = 5.15, 95% CI [1.14, 47.43], p = 0.02). Grand multiparous (≥6 times) demonstrated higher odds of mastitis than primiparous (OR = 2.13, 95% CI [1.61, 4.81], p < 0.001). In addition, cows in advanced lactation stages (≥ 7 months) showed higher odds of mastitis than early stage (OR = 2.65, 95% CI [1.47, 4.78], p < 0.001). Dairy cows categorized as very dirty demonstrated substantially higher odds of mastitis compared to those categorized as not dirty (OR = 85.47, 95% CI [27.18, 268.74], p < 0.001).

**Extrinsic factors.** *Milking and hygienic milk handling practices*: The results are presented in Table 5. Cows milked by hand showed higher mastitis odds than those milked by milking machine (OR = 9.59, 95% CI [1.21, 433.30], p = 0.023). Mastitis occurrence was high in farmers who allowed the calf to suckle before and within milking than whose do not allow calves to suckle (OR = 11.02, 95% CI [1.449, 490.310], p = 0.006). The odds of mastitis were higher in cows that were incompletely milked, where milk was voluntarily left in the udder for calves to suckle after milking compared to cows whose calves were not used (OR = 7.32, 95% CI [2.380, 29.912], p < 0.001). Additionally, mastitis odds increased with prolonged milking durations exceeding 15 minutes compared to cows milked less than 7 minutes.

**Table 3. The prevalence of mastitis at cow side and quarter levels.**

| Categories | Number of examined milking cows/quarters | % Nr of positive cases | 95% CI | Standard Error | Margin of Error |
|---|---|---|---|---|---|
| **Subclinical mastitis** | | | | | |
| Overall cow-side SCM | 411* | 234 (56.9%) | 52.1–61.7 | 0.0244 | 0.0478 |
| Quarter SCM | 1639** | 565 (34.4%) | 32.1-36.7 | 0.0117 | 0.0230 |
| **Clinical mastitis** | | | | | |
| Overall CM | 411* | 13 (3.16%) | 1.47-4.85 | 0.0086 | 0.0169 |
| Change of udder/teats symmetry | 411* | 6 (1.45%) | 0.30-2.61 | 0.0059 | 0.0115 |
| Milk with clots | 411* | 1 (0.24%) | −0.23-0.71 | 0.0024 | 0.0047 |

SCM: Sub-clinical Mastitis; CM: Clinical Mastitis; *Cows; **Quarters.

**Table 4. Multinomial logistic regression results for factors of selected intrinsic and cow related factors versus mastitis (n = 411).**

| Variables | Categorie | Mastitis variation | | OR | 95% CI | P-Value |
|---|---|---|---|---|---|---|
| | | Positive | Negative | | | |
| **Grazing systems** | Open grazing | 138 | 143 | | Ref | |
| | Zero grazing | 96 | 34 | 2.93 | [1.81, 4.76] | <0.001 |
| **Age** | Old (≥7 years) | 67 | 40 | | Ref | |
| | Young (1–6 years) | 167 | 137 | 0.73 | [0.45, 1.17] | 0.175 |
| **Breed** | Cross breed | 221 | 175 | | Ref | |
| | Exotic pure breed | 13 | 2 | 5.15 | [1.14, 47.43] | 0.02 |
| **Parity** | Primiparous | 88 | 98 | | Ref | |
| | Multiparous (2–5 times) | 26 | 16 | 1.82 | [0.53, 4.94] | 0.12 |
| | Grand multiparous (≥6 times) | 120 | 63 | 2.13 | [1.61, 4.81] | <0.001 |
| **Stage of lactation** | Early (1–2 months) | 58 | 70 | | Ref | |
| | Middle (3–6 months) | 107 | 77 | 1.67 | [1.04, 2.69] | 0.033 |
| | Later (≥7 months) | 68 | 31 | 2.65 | [1.47, 4.78] | 0.001 |
| **Presence of teats ulcerations and cracks** | No | 216 | 166 | | Ref | |
| | Yes | 18 | 11 | 1.2576 | [0.54, 3.03] | 0.697 |
| **Cow hygiene** | Not dirty | 63 | 76 | | Ref | |
| | Slightly dirty | 12 | 91 | 0.16 | [0.07, 0.38] | <0.001 |
| | Fairly dirty | 17 | 8 | 2.56 | [0.95, 6.92] | 0.062 |
| | Very dirty | 142 | 2 | 85.47 | [27.18, 268.74] | <0.001 |

Mastitis odds were higher in farms without consistent access to clean water (OR = 3.89, 95% CI [2.50, 6.28], p < 0.001). Mastitis was relatively higher in farmers who applied milking jelly or salve to the udder before milking than those who did not use (OR = 2.30, 95% CI [1.01, 5.41], p = 0.035).

*Mastitis control and farming practices*: The results for mastitis control and farming practices are presented in Table 6. The prevalence of mastitis was higher among farmers who had never conducted screening compared to those who performed daily screening (OR = 5.48, 95% CI [1.64, 18.33], p = 0.002). The odds of mastitis were significantly higher among farmers who did not practice pre-teat dipping (OR = 11.03, 95% CI [1.45, 490.19], p = 0.006) and who did not practice post-teat dipping (OR = 7.97, 95% CI [3.88, 16.39], p < 0.0001). It was also significantly higher among farmers who did not practice dry cow therapy (OR = 13.12, 95% CI [1.82, 556.30], p = 0.003). Mastitis prevalence was relatively higher among farmers without appropriate cowshed or confinement facilities (OR = 12, 95% CI [2.559, 56.463], p = 0.00019), as well as

**Table 5. Multinomial logistic regression results for factors of on-farm milking practices and mastitis (n = 411).**

| Variables | Categories | Mastitis variation | | OR | 95% CI | P-Value |
|---|---|---|---|---|---|---|
| | | Positive | Negative | | | |
| **Milking and milk handling practices** | | | | | | |
| **Milking techniques** | Milking machine | 1 | 7 | | Ref | |
| | Hand milking | 233 | 170 | 9.59 | [1.169, 78.711] | 0.028 |
| **Let the calf suckle before and/or within milking** | No | 1 | 8 | | Ref | |
| | Yes | 233 | 169 | 11.02 | [1.367, 89.022] | 0.014 |
| **Keep/leave teats for calves to suckle after milking** | No | 4 | 20 | | Ref | |
| | Yes | 230 | 157 | 7.32 | [2.456, 21.824] | <0.001 |
| **Time used to milk a cow** | Less than 7 mins | 186 | 155 | | Ref | |
| | Between 8–15 mins | 37 | 21 | 1.47 | [0.825, 2.613] | 0.243 |
| | More than 15 min | 11 | 1 | 9.17 | [1.17, 71.794] | 0.024 |
| **Milking cows while suffering from contagious diseases like diarrhea or typhoid** | No | 220 | 118 | | Ref | |
| | Yes | 14 | 9 | 0.834 | [0.351, 1.985] | 0.854 |
| **Presence of clean water on the farm** | Yes | 83 | 96 | | (Ref) | |
| | No | 179 | 53 | 3.91 | [2.555, 5.972] | <0.001 |
| **Udder preparation before milking** | Clean with cold water only | 160 | 110 | | Ref | |
| | Clean the udder with warm water only | 65 | 45 | 0.99 | [0.633, 1.559] | 1 |
| | Apply milking jelly/salve | 12 | 19 | 0.43 | [0.203, 0.931] | 0.046 |
| **Drying the udder with a tower or cloth before milking** | Yes | 56 | 40 | | Ref | |
| | No | 178 | 137 | 0.93 | [0.584, 1.474] | 0.843 |
| **Milkers' preparation before milking** | Warm water only | 24 | 23 | | Ref | |
| | Warm water and detergent | 59 | 32 | 1.77 | [0.864, 3.615] | 0.167 |
| | Cold water only | 12 | 32 | 0.36 | [0.15, 0.863] | 0.035 |
| | Cold water and detergent | 139 | 101 | 1.32 | [0.705, 2.468] | 0.48 |
| **Who cleans milking utensils/equipment** | Husband | 9 | 2 | | Ref | |
| | Wife | 92 | 61 | 0.34 | [0.07, 1.605] | 0.207 |
| | Child | 15 | 9 | 0.37 | [0.065, 2.112] | 0.453 |
| | Workers/ cow keepers | 118 | 105 | 0.25 | [0.053, 1.182] | 0.117 |
| **Cleaning milking utensils/equipment** | Clean with warm only | 24 | 23 | | Ref | |
| | Clean with cold water only | 49 | 26 | 1.81 | [0.858, 3.8] | 0.169 |
| | Warm water and detergent | 27 | 25 | 1.04 | [0.47, 2.279] | 1 |
| | Cold water and detergent | 134 | 103 | 1.25 | [0.666, 2.334] | 0.596 |
| **Selling milk at the Milk Collection Center (MCC)** | No | 138 | 99 | | Ref | |
| | Yes | 96 | 78 | 0.882 | [0.595, 1.31] | 0.605 |
| **Containers used for milking and milk handling** | Aluminum | 37 | 20 | | Ref | |
| | Plastic containers | 189 | 153 | 0.667 | [0.372, 1.198] | 0.224 |
| | Stainless steel containers | 8 | 4 | 1.08 | [0.289, 4.038] | 1 |

those who lacked proper record-keeping practices (OR = 4.53, 95% CI [2.39, 8.92], p < 0.0001). A higher proportion of farmers treat mastitis themselves.

*Environmental and other external factors:* Within zero-grazing systems, inadequate drainage of cowsheds was significantly associated with higher odds of mastitis (OR=109, 95% CI [26.46, 507.18], p < 0.001). Infrequent bedding change (≥ a month) was also strongly associated with higher odds of mastitis (OR = 6.364, 95% CI [3.381, 12.783], p = 0.014) compared to changing less than once a week. Likewise, mastitis odds were higher in regions where access to public veterinary services

**Table 6. Multinomial logistic regression results for factors of on-farm mastitis control and farming practices versus mastitis (n = 411).**

| Variables | Categories | Mastitis variation | | OR | 95% CI | P-Value |
|---|---|---|---|---|---|---|
| | | Positive | Negative | | | |
| **Mastitis control and farming practices** | | | | | | |
| **Checking for clinical mastitis before milking (teat striping)** | Yes | 140 | 105 | | Ref | |
| | No | 94 | 72 | 1.02 | [0.67, 1.55] | 0.92 |
| **Subclinical mastitis screening frequency.** | Everyday | 4 | 13 | | Ref | |
| | Once a week | 6 | 10 | 1.94 | [0.34, 11.13] | 0.465 |
| | Once a month | 23 | 35 | 2.13 | [0.56, 8.14] | 0.264 |
| | Never done | 201 | 119 | 5.48 | [1.64, 18.33] | 0.002 |
| **Treat mastitic cows** | Yes | 115 | 85 | | Ref | |
| | No | 119 | 92 | 1.05 | [0.69, 1.58] | 0.842 |
| **Who treats the cows (administers the medicine) during mastitis treatment?** | Veterinarians | 58 | 51 | | Ref | |
| | Myself | 174 | 124 | 1.233 | [0.89, 2.19] | 0.147 |
| | Neighbor farmer | 2 | 2 | 0.88 | [0.008, 0.02] | <0.001 |
| **Do you seek a prescription from the veterinarians before treatment?** | Yes | 26 | 22 | | Ref | |
| | No | 208 | 155 | 1.135 | [0.58-2.18] | 0.757 |
| **Pre-teat dipping** | Yes | 1 | 8 | | Ref | |
| | No | 233 | 169 | 11.03 | [1.45, 490.19] | 0.006 |
| **Post- teat dipping** | Yes | 7 | 27 | | Ref | |
| | No | 227 | 110 | 7.97 | [3.88, 16.39] | <0.0001 |
| **Dry cow therapy** | Yes | 1 | 10 | | Ref | |
| | No | 227 | 173 | 13.12 | [1.82, 556.30] | 0.003 |
| **Presence of Cowshed** | Yes | 2 | 12 | | Ref | |
| | No | 78 | 38 | 12 | [2.559, 56.463] | 0.00019 |
| **Record keeping** | Yes | 16 | 45 | | Ref | |
| | No | 216 | 134 | 4.53 | [2.39, 8.92] | <0.0001 |

was limited (OR = 3.45, 95% CI [2.22, 5.38], p<0.001). Furthermore, cows sprayed less than twice a week against vectors like ticks and mosquitoes had significantly higher odds of mastitis (OR = 5.51, 95% CI [2.573, 11.803], p<0.001) (Table 7).

### Gaps associated with mastitis occurrence in the current Rwandan dairy system

A conceptual diagram illustrating gaps in mastitis control was developed from qualitative analysis of farmer interviews and field observations (Fig 1). Based on the findings, key gaps identified include limited farmers' knowledge in animal health management, lack of affordable mastitis control supplies (e.g., disinfectants, California Mastitis Test kits), and inadequate access to clean water. The absence of routine mastitis screening for monitoring prevalence and incidence at the farm level, along with the lack of a national mastitis control strategy, represents a significant gap. Farm biosecurity-related gaps identified in this study include inadequate environmental management, inappropriate housing, poor control of mastitis-causing pathogens, and improper milking practices.

Furthermore, this study identified the dominance of informal milk market channels, coupled with unfair milk pricing, poor milk quality along the value chain, and weak enforcement of milk marketing policies as factors that exacerbate the increased risk of mastitis at the farm level.

## Discussion

The prevalence, risk factors and gaps in the Rwandan dairy system were deeply investigated by this study. By integrating both mastitis screening and individual questionnaire, this study provides useful information to design appropriate mastitis control programs in Rwanda and similar grazing systems.

**Table 7. Multinomial logistic regression results for factors of on-farm mastitis environmental and other external influencing factors versus mastitis (n = 411).**

| Variables | Categories | Mastitis variation | | OR | 95% CI | P-Value |
|---|---|---|---|---|---|---|
| | | Positive | Negative | | | |
| **Environmental management and other external factors** | | | | | | |
| **Does your cowshed have a proper slope and good drainage?** | Yes | 8 | 46 | | Ref | |
| | No | 72 | 4 | 109 | [26.46, 507.18] | < 0.001 |
| **How frequently do you change the bedding?** | Everyday | 1 | 3 | | Ref | |
| | Less one week | 2 | 6 | 1 | [0.037, 26.951] | 0.134 |
| | Between 1 week- two weeks | 7 | 8 | 2.625 | [0.390, 34.425] | 0.073 |
| | Once a month or over | 70 | 33 | 6.364 | [3.381, 12.783] | 0.014 |
| **Is the milking area separated from the graz-ing area?** | Yes | 151 | 119 | | Ref | |
| | No | 3 | 8 | 0.3 | [0.05, 1.26] | 0.071 |
| **How long does it take for a cow to lay down after milking?** | More than an hour | 63 | 53 | | Ref | |
| | Between 30 mins – 1 hour | 71 | 111 | 0.537 | [0.322, 0.896] | 0.018 |
| | Less than 30 mins | 24 | 9 | 2.242 | [1.234, 4.058] | 0.008 |
| | Don't know | 36 | 44 | 1.378 | [0.415, 1.147] | 0.162 |
| **Are private veterinarians available and accessible to you?** | Yes | 136 | 109 | | Ref | |
| | No | 98 | 69 | 0.721 | [0.749, 1.303] | 0.545 |
| **Are public veterinarians available and accessible to you?** | Yes | 66 | 102 | | Ref | |
| | No | 168 | 75 | 3.45 | [2.22, 5.38] | < 0.001 |
| **Do you have access to mastitis-related training?** | Yes | 15 | 97 | | Ref | |
| | No | 216 | 83 | 16.82 | [9.21, 30.58] | < 0.001 |
| **Do you use molasses in your feeding routine?** | Yes | 1 | 6 | | Ref | |
| | No | 233 | 171 | 233 | [1.027, 376.85] | 0.046 |
| **How frequently do you spray against vec-tors such as ticks and mosquitoes?** | Twice a week | 58 | 64 | | Ref | |
| | Once a week | 115 | 102 | 1.102 | [0.697, 1.745] | 0.676 |
| | Over a week | 60 | 12 | 5.51 | [2.573, 11.803] | < 0.001 |

## Mastitis prevalence

The overall cow-level subclinical mastitis (SCM) of 60.06% is comparable to previous studies in Rwandan [18], but slightly higher than 51.8% [7] and 50.4% [19] while relatively lower than 76.2% [20]. Notably, our findings are lower than 73.1% SCM in Kenya [21] and 68.6% SCM in Uganda [22] but higher than 32.21% SCM in Ethiopia [23] and 52.3% in Egypt [6]. The observed clinical mastitis prevalence of 3.14% was lower than 32.8% [11] and 12.59% [23] both in Ethiopia, 6.8% in Kenya [21] and 11.6% in Switzerland [1]. These variations may stem from differences in farming practices, genetic resistance, and environmental conditions, underscoring the need for context-specific mastitis control strategies. The high prevalence rates of subclinical mastitis compared to clinical mastitis may be attributed to negligence towards subclinical mastitis and the limited knowledge of smallholder farmers regarding silent cases of mastitis. As highlighted by [24], a mastitis prevalence of 40% or higher on a farm is deemed alarming to producers. This observation underscores mastitis severity within Rwanda's dairy industry, emphasizing the urgent need to address milk quality and production concerns.

## Intrinsic and cow related predisposing factors

The correlation between cow dirtiness and elevated mastitis occurrence corroborate with findings by [1,6,23] emphasizing the critical role of environmental hygiene in mastitis prevention. The observed 2.65 times mastitis odds higher in late lactation

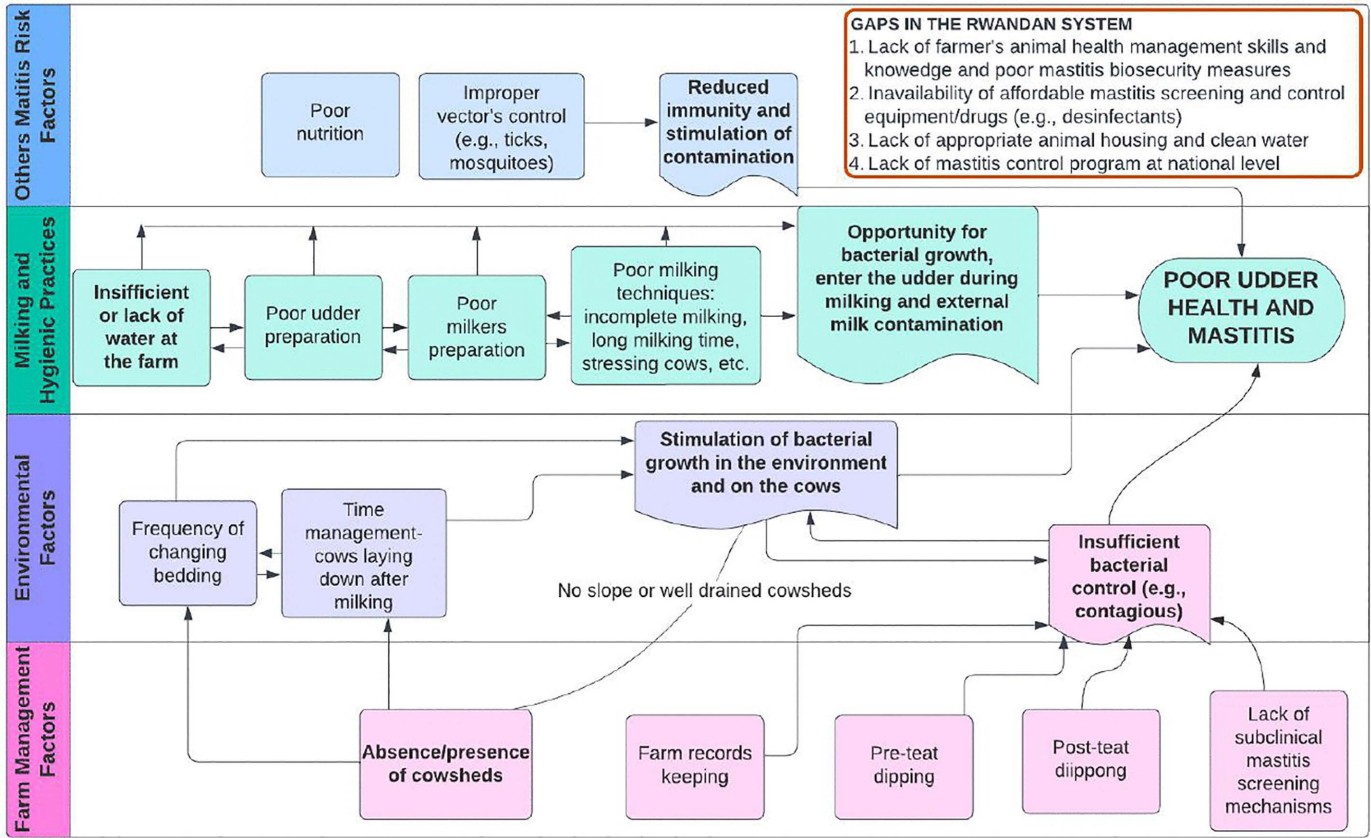

**Fig 1. Gaps associated with mastitis control in the current Rwandan dairy system.**

compared to early stages is consistent with findings by [7,20,25]. However, [23] reported a higher mastitis incidence in early lactation stages. These differences may arise from genetic factors such as breed resistance, udder size, and heritable traits, as well as the timing of mastitis onset, frequently noted towards the end of the preceding dry period or during calving [25]. Mastitis increase in grand multiparous (≥6 times) compared to primiparous is in agreement with [8,20,23]. This trend could be linked to the gradual decline in the body's immune system, repeated exposure to milking, and anatomical changes in the udder and teats [23]. Additionally, the association of mastitis with breed could be attributed to the significant genetic improvement through artificial inseminations in Rwanda leading to high producing dairy cattle breeds prone to mastitis infections [3].

While age is typically regarded as a significant mastitis factor [23], its association with mastitis prevalence was not significant in this study. This is possibly due to inadequate record-keeping also observed in this study leading to potential inaccuracies, particularly for cows not born into those herds. Furthermore, the absence of a correlation between teat alterations and cracks and mastitis may be attributed to the geographical context of this study, which was conducted in North-Western zones characterized by less forest and savannah vegetation protecting teats and udders from significant damage.

## Extrinsic factors and gaps

This study underscores the multifaceted nature of mastitis prevalence, emphasizing its strong correlation with farm management practices, milking procedures, environmental conditions, and external factors. Zero-grazing systems showed a 2.93-times higher mastitis risk than open grazing, largely due to inadequate housing, poor drainage, and infrequent

bedding changes. In the Rwanda's integrated livestock-crop systems [14], farmers often store manure in animal's housing areas to enhance composting, inadvertently creating ideal conditions for bacterial proliferation. While beneficial for soil fertility, this practice significantly increases mastitis risk, underlining the need for improved manure management strategies in zero-grazing systems. Environmental factors in pastoral systems have also been highlighted by researchers in the region including Uganda [22]; Kenya [21]; Ethiopia [23].

Poorly drained milking areas coupled with suboptimal milking practices (e.g., incomplete milking and poor hands and udder wash, etc.), further exacerbate the risk of bacterial growth and contamination. This is potentially linked to pathogens such as *Bacillus* spp., *S. aureus and Salmonella spp.* previously reported in milking cows in Rwanda [10,19], Kenya [21] and Uganda [22]. In addition, the observed poor feeding practices such as nonuse of molasses, may contribute to lower lactose levels and insufficient energy intake, increasing the risk of mastitis occurrence [26]. The observed unexpected associations with vector control practices indicate the potential stress-induced immunosuppression disrupting the delicate balance of homeostasis further predisposing milking cows to inflammatory conditions in the mammary gland.

The association between mastitis occurrence and farm management practices has been consistently highlighted by researchers worldwide [1,6,10,23,27]. Inadequate housing, poor record-keeping, and low farmer literacy were significantly noticed in the current study. The absence of key preventive measures such as use of teat dipping and dry cow therapy contribute to bacterial growth increasing mastitis risk. In addition, the observed frequent use of antibiotics to manage mastitis in this study contributes to antimicrobial resistance (AMR) posing risks to both animal and human health. This is evidenced by identified resistant strains of *S. aureus* and other mastitis pathogens in the Rwandan dairy chain [28] emphasizing the need to integrating mastitis control into national AMR surveillance programs. Though the overall sample size was adequate, some odds ratios showed wide confidence intervals, likely due to low numbers of observations within certain subgroups.

## Conclusion and recommendations

This study is the first to systematically evaluate the prevalence and risk factors of bovine mastitis across distinct grazing systems in Rwanda using a stratified design that links individual cow health outcomes with farm-level management practices. By pairing clinical examinations and mastitis screening with structured interviews of the corresponding cow owners, this study provides a unique dataset that bridges the gap between animal-level and farm-level risk factors.

By identifying critical management gaps, these findings offer practical interventions to improve milk quality, farm profitability, and public health in Rwanda and similar LMIC settings. The observed high cow-level mastitis prevalence of 60.06% highlights the urgent need for targeted interventions. Key contributing factors include inadequate farmer knowledge, poor housing conditions, limited access to clean water, improper milking practices, and poor environmental hygiene. Strengthening farmer training programs, improving farm infrastructure, and implementing effective mastitis control strategies are critical for mitigating mastitis risks. We strongly recommend a collaborative approach among stakeholders in the dairy value chain to establish a nationwide mastitis control program, integrating bacterial control measures, improved water access, and robust surveillance systems to enhance milk quality and dairy farm resilience in Rwanda.

## Supporting information

**S1 Dataset. Anonymized dataset used for statistical analysis of mastitis and associated risk factors.**
(XLSX)

**S2 Appendix. Mastitis screening tool and intrinsic and cow-related factors questionnaire.**
(DOCX)

**S3 Appendix. Questionnaire assessing extrinsic risk factors related to mastitis occurrence.**
(DOCX)

   

## Acknowledgments

We express our gratitude to the leadership of Nyabihu and Musanze districts for their approvals and collaboration. We sincerely acknowledge Benjamin Shumbusho from the University of Rwanda for providing field oversight and the University of Illinois Urbana-Champaign for project supervision. Furthermore, we appreciate the farmers and milk collection centers in Nyabihu and Musanze districts for their valuable time and collaboration during this research.

## Author contributions

**Conceptualization:** Blaise Iraguha.

**Data curation:** Blaise Iraguha, Methode Ngabo Gasana.

**Formal analysis:** Blaise Iraguha.

**Investigation:** Blaise Iraguha, Methode Ngabo Gasana.

**Methodology:** Blaise Iraguha, Jean Pierre M. Mpatswenumugabo.

**Supervision:** Jean Pierre M. Mpatswenumugabo.

**Writing – original draft:** Blaise Iraguha, Methode Ngabo Gasana, Jean Pierre M. Mpatswenumugabo.

**Writing – review & editing:** Blaise Iraguha.

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
