## [Decision Letter · Decision Letter 0]

29 Sep 2025

PONE-D-25-38271Bovine Mastitis Epidemiology: Prevalence, Risk Factors, Control Program Gaps and Biosecurity Recommendations to Improve Animal Health in the Rwandan Smallholder Dairy Farms.PLOS ONE

Dear Dr. Iraguha,

Thank you for submitting your manuscript to PLOS ONE. After careful consideration, we feel that it has merit but does not fully meet PLOS ONE’s publication criteria as it currently stands. Therefore, we invite you to submit a revised version of the manuscript that addresses the points raised during the review process.

We look forward to receiving your revised manuscript.

Kind regards,

Mourad Mahmoud

Academic Editor

PLOS ONE

Journal Requirements:

2. In the ethics statement in the Methods, you have specified that verbal consent was obtained. Please provide additional details regarding how this consent was documented and witnessed, and state whether this was approved by the IRB

4, When completing the data availability statement of the submission form, you indicated that you will make your data available on acceptance. We strongly recommend all authors decide on a data sharing plan before acceptance, as the process can be lengthy and hold up publication timelines. Please note that, though access restrictions are acceptable now, your entire data will need to be made freely accessible if your manuscript is accepted for publication. This policy applies to all data except where public deposition would breach compliance with the protocol approved by your research ethics board. If you are unable to adhere to our open data policy, please kindly revise your statement to explain your reasoning and we will seek the editor's input on an exemption. Please be assured that, once you have provided your new statement, the assessment of your exemption will not hold up the peer review process.

6. We note that Figure 1 in your submission contain [map/satellite] images which may be copyrighted. All PLOS content is published under the Creative Commons Attribution License (CC BY 4.0), which means that the manuscript, images, and Supporting Information files will be freely available online, and any third party is permitted to access, download, copy, distribute, and use these materials in any way, even commercially, with proper attribution. For these reasons, we cannot publish previously copyrighted maps or satellite images created using proprietary data, such as Google software (Google Maps, Street View, and Earth). For more information, see our copyright guidelines: http://journals.plos.org/plosone/s/licenses-and-copyright.

Reviewers' comments:

Reviewer's Responses to Questions

**Comments to the Author**

1. Is the manuscript technically sound, and do the data support the conclusions?

Reviewer #1: Yes

Reviewer #2: Yes

2. Has the statistical analysis been performed appropriately and rigorously? 

Reviewer #1: Yes

Reviewer #2: Yes

3. Have the authors made all data underlying the findings in their manuscript fully available?

Reviewer #1: Yes

Reviewer #2: Yes

4. Is the manuscript presented in an intelligible fashion and written in standard English?

Reviewer #1: Yes

Reviewer #2: Yes

5. Review Comments to the Author

Reviewer #1: The manuscript is well-designed, relevant, and makes a valuable contribution to understanding bovine mastitis in Rwandan smallholder dairy systems. The methodology and analyses are appropriate, and the findings are clearly presented with practical recommendations. Only minor issues require attention, including consistency in terminology, clarity of expression, table/figure formatting, and harmonization of the data availability statement. No concerns regarding research ethics, dual publication, or publication ethics were identified. I recommend acceptance after minor revision. Kindly refer to the reviewer attachment.

Reviewer #2: This report is well-written and describes in details the situation of dairy production in Rwanda, regarding in particular the risk factors linked to subclinical and clinical mastitis.

My only comment is for table 5 which shows inconsistencies between OR and reported positive and negative cases: for instance "Uder preparation before milking" shows an OR of 2.3 for "Apply milking jelly/salve" while it shows reduced numbers of positive cases... either the OR reported is wrong or the authors have inverted positive/negative cases. Same applies for other lines in this table. Other tables seem OK.

6. PLOS authors have the option to publish the peer review history of their article (what does this mean? ). If published, this will include your full peer review and any attached files.

**Do you want your identity to be public for this peer review?** For information about this choice, including consent withdrawal, please see our Privacy Policy .

Reviewer #1: **Yes:** Mohammad Sabri Abdul Rahman

Reviewer #2: No

---

## [Author Response · Author response to Decision Letter 1]

15 Nov 2025

Editorial Comments

Comment 1: Please ensure that your manuscript meets PLOS ONE's style requirements, including those for file naming.

Response: We reviewed the journal’s style guidelines and adjusted the manuscript accordingly. Please refer to the final revised “Manuscript”.

Comment 2: In the ethics statement in the Methods, you have specified that verbal consent was obtained. Please provide additional details regarding how this consent was documented and witnessed, and state whether this was approved by the IRB.

Response: Formal IRB review was not required because the study did not involve the collection of human or animal biological samples or any sensitive personal data. Instead, ethical approval was obtained from the relevant administrative authorities at the district level in Nyabihu and Musanze, Rwanda, in accordance with Rwanda’s national research ethics guidelines. The study’s objective and procedures were explained to all participating farmers. As many participants were unable to read or write, verbal informed consent was obtained prior to each animal clinical assessment and interview, and the process was witnessed and documented by a local veterinary or extension officer. The revised Materials and Methods – Ethical Considerations section now includes this clarification and confirms compliance with the Declaration of Helsinki principles. (Lines 107–115).

Comment 3: We note that your Data Availability Statement is currently as follows: 'All relevant data are within the manuscript and its Supporting Information files.'

Response: Revised for conciseness and consistency with PLOS ONE’s requirements. Additional Excel data and the questionnaire used for data collection have been uploaded as a zip file in supporting information (Line 410).

Comment 4: Data sharing policy: All research data must be made freely accessible upon acceptance unless ethical restrictions apply.

Response: The full dataset has been submitted as supporting information.

Comment 5: Ethics statement: The ethics statement must appear only in the Methods section of the manuscript.

Response: The ethics statement now appears only in the Methods section (Lines 107–115).

Comment 6: Copyright compliance: Figures using copyrighted map or satellite images (e.g., Google Maps) cannot be published unless permission is granted.

Response: Permission was requested from ESRI, as we used ArcGIS to generate the map; however, we did not receive authorization for the copyrighted material used in Figure 1. Therefore, Figure 1 was removed from the manuscript, and the relevant geographical information was retained in the text.

Comment 7: Reviewer-suggested citations: You should evaluate any papers suggested by reviewers and cite them only if relevant.

Response: All suggested references were evaluated, and only relevant ones were addressed.

Comment 8: Reference list accuracy: Review and correct your reference list, remove or justify any retracted papers, and note all reference changes.

Response: The reference list was reviewed and corrected. No retracted papers are cited; capitalization and URLs were standardized.

Reviewer Comments

Comment 1: Some sentences are long and could be made more concise (particularly in the Introduction and Discussion).

Response: Long sentences in the entire manuscript were shortened for better readability (e.g., Abstract Lines 41–43; Introduction Lines 55–57, Discussion: 329-335).

Comment 2: Use uniform terms for cow hygiene (e.g., 'very dirty,' 'dirty’) currently mixed with 'muddy filled.'

Response: A uniform hygiene scoring system ('Not dirty,' 'Slightly dirty,' 'Fairly dirty,' 'Very dirty') was applied (Lines 174–177) and removes confusion. The term 'mastitis control variants' was revised to “mastitis control supplies” (e.g., disinfectants such as iodine or chlorine dioxide)' (Lines 37–38).

Comment 3: Some odds ratios have extremely wide confidence intervals (e.g., OR = 11.02; 95% CI: 1.449–490.310). Mention this limitation briefly in the Discussion.

Response: Wide confidence intervals were acknowledged as a limitation in the Discussion (Lines 367–369).

Comment 4: Ensure tables are formatted consistently (some captions are embedded mid-text).

Response: All tables were reformatted for consistency, and captions properly aligned. Figure 1 was removed; Figure 2 was relabeled and cited correctly.

Comment 5: In the main manuscript, the Data Availability statement says 'Data will be made available on request.' Please ensure consistency with PLOS ONE’s data-sharing policy.

Response: The Data Availability Statement was unified to read: 'All relevant data are within the manuscript and its Supporting Information.' The dataset has been attached.

Comment 6: Check reference formatting for consistency (e.g., spacing, capitalization, and journal abbreviations).

Response: Reference formatting, capitalization, spacing, and URLs were checked and corrected.

Comment 7: A few minor grammatical issues (e.g., 'mastitis key risk factors' → 'key mastitis risk factors').

Response: Minor grammatical issues were corrected (e.g., 'key mastitis risk factors,' 'poor farm biosecurity,' and 'occasional deaths') (Lines 21, 38, 63).

We believe these revisions have addressed all comments and significantly improved the manuscript. We sincerely thank the editor and reviewers for their valuable input and consideration.

---

## [Editor Report · Decision Letter 1]

2 Dec 2025

Bovine mastitis epidemiology: prevalence, risk factors, control program gaps and biosecurity recommendations to improve animal health in the Rwandan smallholder dairy farms.

PONE-D-25-38271R1

Dear Dr. Iraguha,

We’re pleased to inform you that your manuscript has been judged scientifically suitable for publication and will be formally accepted for publication once it meets all outstanding technical requirements.

Kind regards,

Mourad Mahmoud

Academic Editor

PLOS ONE

---

## [Editor Report · Acceptance letter]

PONE-D-25-38271R1

PLOS One

Dear Dr. Iraguha,

I'm pleased to inform you that your manuscript has been deemed suitable for publication in PLOS One. Congratulations! Your manuscript is now being handed over to our production team.

Kind regards,

on behalf of

Dr. Mourad Mahmoud

Academic Editor

PLOS One